

# *Proceraea exoryxae* sp. nov. (Annelida, Syllidae, Autolytinae), the first known polychaete miner tunneling into the tunic of an ascidian

Daniel Martin[1], Arne Nygren[2] and Edwin Cruz-Rivera[3]

[1] Centre d'Estudis Avançats de Blanes (CEAB–CSIC), Girona, Spain
[2] Sjöfartsmuseet Akvariet, Göteborg, Sweden
[3] Department of Biological Sciences, University of the Virgin Islands, St. Thomas, Virgin Islands, USA

## ABSTRACT

While studying organisms living in association with the solitary tunicate *Phallusia nigra* (Ascidiacea, Ascidiidae) from a shallow fringing reef at Zeytouna Beach (Egyptian Red Sea), one of the collected ascidians showed peculiar perforations on its tunic. Once dissected, the perforations revealed to be the openings of a network of galleries excavated in the inner tunic (atrium) by at least six individuals of a polychaetous annelid. The worms belonged to the Autolytinae (Syllidae), a subfamily that is well known to include specialized predators and/or symbionts, mostly associated with cnidarians. The Red Sea worms are here described as *Proceraea exoryxae* sp. nov., which are anatomically distinguished by the combination of simple chaetae only in anterior chaetigers, and a unique trepan with 33 teeth in one outer ring where one large tooth alternates with one medium-sized tricuspid tooth, and one inner ring with small teeth located just behind the large teeth. Male and female epitokes were found together with atokous individuals within galleries. *Proceraea exoryxae* sp. nov. constitutes the first known miner in the Autolytinae and the second species in this taxon known to live symbiotically with ascidians. The implications of finding this specialized parasite are discussed considering that *Phallusia nigra* has been introduced worldwide, in tropical and sub-tropical ecosystems, where it has the potential of becoming invasive.

## INTRODUCTION

There are approximately 11,840 polychaete annelids known, spanning a remarkable array of habitats, ecological niches, and trophic modes (*Read & Fauchald, 2016*). Among these, symbiotic species (sensu *Castro, 2015*) span at least 28 different families (*Martin & Britayev, 1998*). These symbiotic interactions, in general, are poorly understood, but cases of inquilinism, commensalism, mutualism and parasitism have been documented. Interestingly, parasitism seems to be among the least common modes of life for polychaetes (<0.5% of known species, spread among 13 families), most of them being found within the Spionidae and most often being shell borers

Corresponding author
Edwin Cruz-Rivera,
edwin.cruzrivera@uvi.edu

(*Martin & Britayev, 1998*). Several reports of associations with tunicates (Phylum Chordata) are available (*Okada, 1935*; *Spooner, Wilson & Trebble, 1957*; *Fiore & Jutte, 2010*), but the polychaetes have not been identified in some of these instances (*Illg, 1958*; *Monniot, 1990*). There are few details known for these associations although consumption of the ascidian host has been reported in one case (*Spooner, Wilson & Trebble, 1957*).

*Phallusia nigra Savigny, 1816*, is a solitary ascidian that has been introduced into tropical and subtropical ecosystems worldwide since it was originally discovered in the Red Sea (*Shenkar, 2012*; *Vandepas et al., 2015*; *Zhan et al., 2015*). The ascidian hosts a remarkable array of crustacean symbionts, including amphipods and at least eight confirmed copepod species (*Kim et al., 2016*). During studies on the ecology of *Phallusia nigra* and its associated fauna in the Egyptian coast of the Red Sea, one of the collected specimens showed various perforations on its tunic. Upon dissection, we discovered a network of excavated galleries resembling the habit of some leaf-mining herbivores in terrestrial and marine habitats (*Brearley & Walker, 1995*; *Connor & Taverner, 1997*; *Sinclair & Hughes, 2010*; *Mejaes, Poore & Thiel, 2015*). The galleries were inhabited by several specimens of a small polychaete species belonging to the subfamily Autolytinae (Annelida, Syllidae). Although some bivalves and crustaceans have been reported to live within ascidian tunics (*Lambert, 2005*; *McClintock et al., 2009*; *Morton & Dinesen, 2011*; *Cañete & Rocha, 2013*), no previous reports of annelids exhibiting a similar habit are known (*Lambert, 2005*; *Monniot, 1990*).

The Autolytinae are small free-living polychaetes, ranging from 1 to 60 mm long and from 0.1 to 1.2 mm wide. They are distributed worldwide and inhabit shallow waters, mostly restricted to the continental shelf. They often live in a more or less intimate association with sedentary invertebrates on which they supposedly feed, such as cnidarians (usually hydroids), but also bryozoans, sponges and tunicates (*Okada, 1928*; *Hamond, 1969*; *Fauchald & Jumars, 1979*; *Genzano & San Martín, 2002*; *Nygren, 2004*; *Nygren & Pleijel, 2007*; *Martin et al., 2015*). Autolytines are commonly found living inside thin, semi-hyaline tubes, either made in association with the host or secreted by the worms and attached directly to the colonial animals with which they associate (*Gidholm, 1967*; *Fischer, Mewes & Franke, 1992*; *Genzano & San Martín, 2002*).

Autolytinae constitute a phylogenetically well-delineated group of polychaetes in the family Syllidae (*Aguado & San Martín, 2009*), characterized by a sinuous pharynx, absence of ventral cirri, presence of simple bayonet-type dorsal chaetae, and reproduction with dimorphic sexes (*Franke, 1999*; *Nygren & Sundberg, 2003*; *Nygren, 2004*). Since the comprehensive revision by *Nygren (2004)*, numerous new species have been described (*Çinar & Gambi, 2005*; *Nygren & Pleijel, 2007*; *Lucas, San Martín & Sikorski, 2010*; *Nygren et al., 2010*; *Álvarez-Campos, San Martín & Piotrowiski, 2014*; *Çinar, 2015*; *Dietrich et al., 2015*; *Martin et al., 2015*; *Aguirre, San Martín & Álvarez-Campos, 2016*). Currently, the subfamily comprises 180 nominal species, of which 112 are considered valid and distributed among 13 recognized genera (*Nygren & Pleijel, 2007*; *Nygren et al., 2010*; *Rivolta, San Martín & Sikorski, 2016*). Among them, *Proceraea* Ehlers, 1,864 contains 28 species (*Nygren, 2004*; *Nygren et al., 2010*; *Martin et al., 2015*).

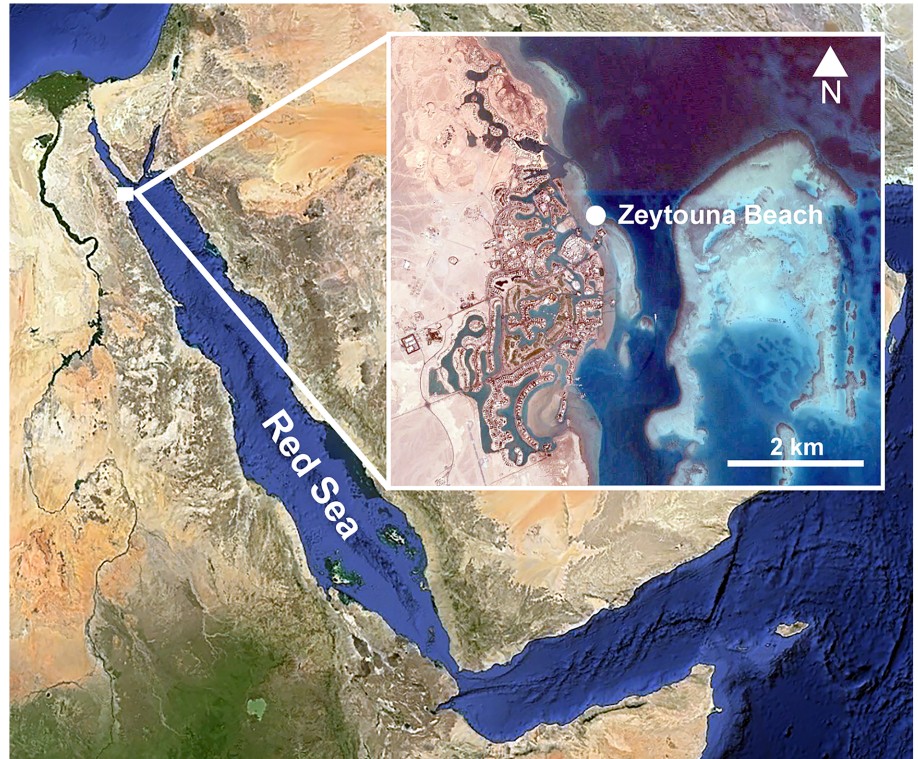

**Figure 1** **Location of the sampling site.** Zeytouna Beach, Egyptian coast of the Red Sea. Maps are from Google Earth Pro, © 2016 DigitalGlobe, © 2016 CNS/Astrium.

It is in *Proceraea* that we place the new species herein described, which occurs inside galleries excavated in the tunic of *Phallusia nigra* and is, thus, the first known miner autolytine. This finding led us to discuss the current knowledge on symbioses involving autolytines, as well as the possible ecological implications of the symbiotic relationship between the polychaete and its host ascidian.

## MATERIALS AND METHODS

Individuals of *Phallusia nigra* were collected by SCUBA from the shallow fringing reef at Zeytouna Beach, on the Egyptian Red Sea (27°24′09.2″N 33°41′08.5″E; Fig. 1) under the auspices of the John D. Gerhart Field Station in El Gouna (American University in Cairo), with permission from the management of Zeytouna Beach. All ascidians were collected on October 7, 2010 at 3–7 m depth and brought to the El Gouna Field Station. In the laboratory, the specimens of *Phallusia nigra* ($N = 50$) were dissected with an incision around the entire periphery of the tunic, and the visceral mass and the pharyngeal sac were removed (Fig. 2A). All of them were inspected for associated animals. Ascidian masses and any abnormalities or damage on the hosts were recorded. Dissected hosts and symbionts from the atrial cavity were photographed with a digital camera equipped with a macro lens.

The entire tunic of the infested ascidian specimen was placed in formaldehyde for a few seconds. Then, the galleries were cut with an angular-tipped scalpel through the atrial

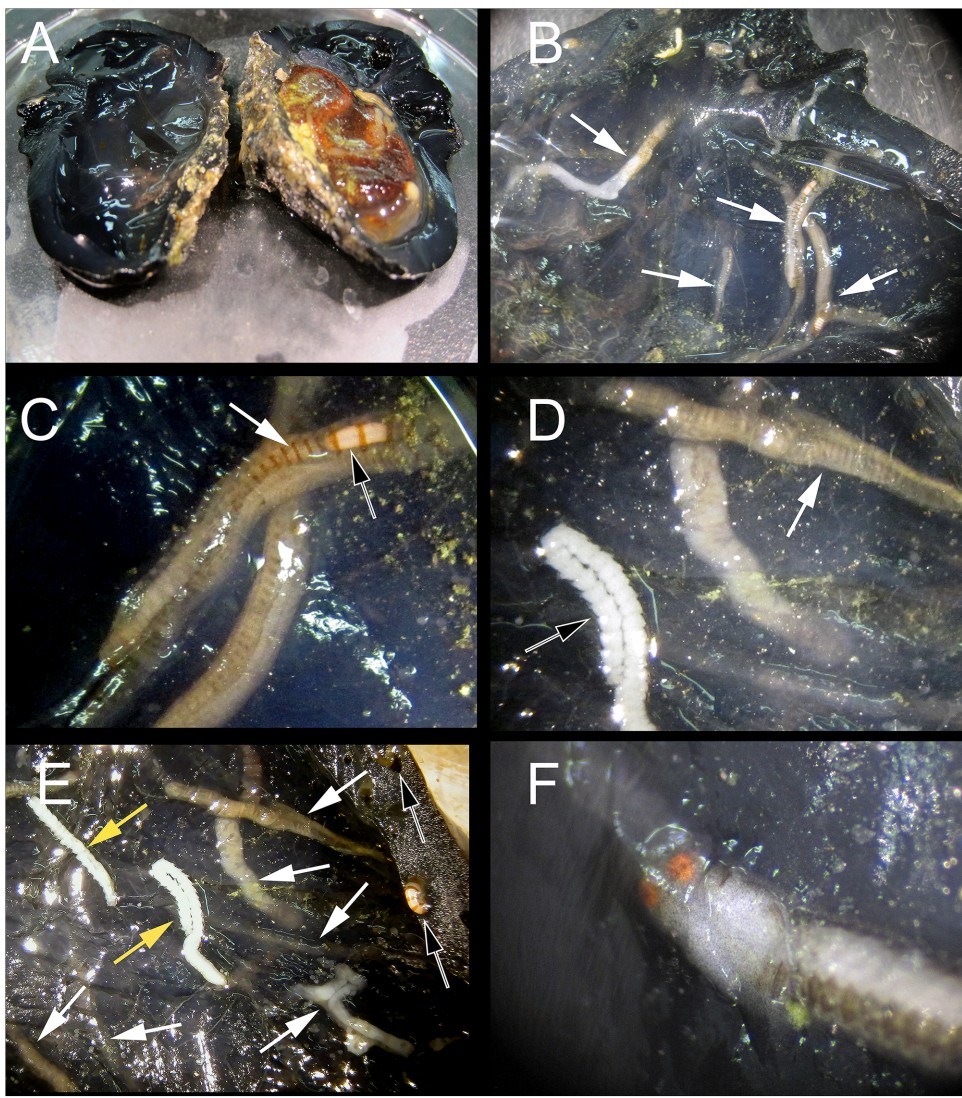

**Figure 2 Host dissection and location of mining polychaetes.** (A) An uninfected *Phallusia nigra* dissected to show normal atrial surface (left) and internal organs. (B) Inner atrial surface showing the presence of several atokous forms inside their galleries (white arrows). (C) Detail of the color of the anterior region of an atokous form; white arrow: position of pharynx; black arrow: position of pro-ventricle. (D) Detail of color of the mid-body of an atokous form (white arrow) and the posterior region of a male epitokous form (black arrow). (E) Inner atrial surface showing the presence of atokous (white arrows) and male epitokous (yellow arrows) forms inside their galleries, as well as part of the banded body of an atokous form protruding from an external tunic opening and other empty tunic openings (black arrows). (F) Close-up view of the head of a male epitoke in its gallery (specimen not preserved).

surface of the tunic to extract mining autolytines with the help of fine forceps. However, their body was very delicate and all of them broke during extraction. In fact, some stolons were completely destroyed in the process and it was not possible to save them for taxonomic studies. All obtained specimens were fixed and preserved in a 4% formalin-seawater solution and transferred to 70% ethanol prior to observations.

Light microscope photos were taken with a Canon EOS 5D Mark II connected to either a Zeiss KF2 triocular microscope via a LM-Scope TUST42C coupler, or a Canon EF

65 mm macro lens with one to five times magnification. For scanning electron microscope (SEM) observations, the 70% ethanol preserved materials were prepared using standard SEM procedures (*Martin et al., 2003*). Prior to run the SEM procedures to observe the trepan, this structure was carefully dissected and as much as possible cleaned from the external muscular tissue layer. Images were taken in a Hitachi TM3000 TABLETOP microscope at the SEM service of the CEAB–CSIC.

The electronic version of this article in portable document format (PDF) will represent a published work according to the International Commission on Zoological Nomenclature (ICZN), and hence the new names contained in the electronic version are effectively published under that Code from the electronic edition alone. This published work and the nomenclatural acts it contains have been registered in ZooBank, the online registration system for the ICZN. The ZooBank LSIDs (life science identifiers) can be resolved and the associated information viewed through any standard web browser by appending the LSID to the prefix http://zoobank.org/. The LSID for this publication is: urn:lsid:zoobank.org:pub:685CB1C2-CB5B-4A87-9CD7-C04BFFDE03B4. The online version of this work is archived and available from the following digital repositories: PeerJ, PubMed Central and CLOCKSS. Specimen vouchers were deposited at the Museo Nacional de Ciencias Naturales of Madrid, Spain (MNCN).

# RESULTS

## Taxonomic account

Phylum ANNELIDA Lamarck, 1809

Subclass ERRANTIA Audouin & Milne-Edwards, 1832

Order PHYLLODOCIDA Dales, 1962

Suborder NEREIDIFORMIA

Family SYLLIDAE Grube, 1850

Subfamily AUTOLYTINAE Langerhans, 1879

Tribe PROCERINI *Nygren, 2004*

Genus *Proceraea* Ehlers, 1864

*Proceraea exoryxae* sp. nov.

LSID. urn:lsid:zoobank.org:act:34373CE6-A0D4-488D-B4A5-12CF4E103504

(Figs. 2–7)

**Type material:** Holotype. MNCN 16.01/17717: atokous anterior fragment, Zeytouna Beach, Egyptian Red Sea, 27°24′09.2″N 33°41′08.5″E, October 7 2010, 3–7 m depth, E. Cruz-Rivera coll.; fixed in 4% formalin seawater, preserved in 70% ethanol. Paratypes. MNCN 16.01/17718: atokous anterior fragment, pharynx dissected; MNCN 16.01/17719: atokous specimen, anterior fragment (up to chaetiger 10) prepared for SEM, mid-body segments and dissected proventricle preserved in 70% ethanol; MNCN 16.01/17720: atokous anterior fragment, pharynx dissected; MNCN 16.01/17721: male stolon, anterior fragment; MNCN 16.01/17722: female

stolon, anterior fragment; MNCN 16.01/17723: atokous mid-body fragments. MNCN 16.01/17724: atokous posterior fragments. Collection details for all other types deposited are the same as for holotype.

**Diagnosis:** *Proceraea* with simple chaetae in anterior chaetigers, and a trepan with 33 teeth with one outer ring where one large tooth alternates with one medium-sized tricuspid tooth, and one inner ring with small teeth located just behind the large teeth.

**Description:** All observations are from preserved specimens if not otherwise stated. Length 3–10.5 mm for 10–68 chaetigers in four anterior fragments, 3–14.5 mm for 19–90 chaetigers in nine median fragments, and 6.5–22 mm for 50–125 chaetigers in three posterior fragments. Width of anterior fragments, excluding parapodial lobes, c. 0.4 mm. Live individuals dorsally with light brown transverse stripes, one per segment, not known whether these are inter- or intrasegmental, or if there is any other additional coloration (Figs. 2B–2E); proventricle white (Figs. 2B and 2C). Formalin preserved specimens without any sign of coloration.

Body shape, excluding parapodial lobes, cylindrical in transection, ventrally flattened. Body long and slender, with slowly tapering end. Nuchal organs ciliated. Prostomium rounded rectangular (Figs. 3A and 3C). Four eyes with lenses, anterior pair larger, confluent in dorsal view, eye spots absent (Fig. 3C). Palps in dorsal view projecting c. half of prostomial length, fused (Figs. 3A and 3B).

Nuchal organs extending to median part of chaetiger 1 (Fig. 3A(A1)). Prostomium with three antennae, median antenna inserted medially on prostomium, lateral antennae on anterior margin. Median antenna reaching chaetiger 8–10, lateral antennae about half as long as median antenna. Tentacular cirri two pairs. Dorsal tentacular cirri about two third as long as median antenna, ventral tentacular cirri about half as long as dorsal tentacular cirri. First dorsal cirri about as long as median antenna, second dorsal cirri as long as ventral tentacular cirri. From chaetiger 3 to chaetiger 20–25, cirri alternate indistinctly in length, shorter cirri slightly shorter and longer cirri equal or slightly longer than body width excluding parapodial lobes (Figs. 3A and 3B), dorsal cirri in more posterior chaetigers more or less equal in length, c. half of body width excluding parapodial lobes; anal cirri as long as half body width, excluding parapodial lobes at level of proventricle.

Cirrophores on tentacular cirri, first and second dorsal cirri (Fig. 3A), otherwise absent. Antennae, tentacular cirri, dorsal cirri, and anal cirri cylindrical. Parapodial lobes rounded. Aciculae 2–3 in anterior chaetigers, 1–2 in median and posterior chaetigers, straight, with a round, swollen distal end (Fig. 4G).

Chaetal fascicle with 9–12 chaetae in anterior chaetigers (Fig. 4A), 4–10 in median and posterior chaetigers. Chaetiger 1–5 with simple chaetae only (Figs. 4B and 5A–5D), chaetiger 6 with simple chaetae only (n = 3), or with single compound chaeta in addition to the simple chaetae (n = 1). From chaetiger 7 to between chaetiger 10–13 with an increasing proportion of compound chaetae (Fig. 4A). Except for the single, thick, distally denticulated bayonet chaeta (Fig. 5E), starting at the earliest in chaetiger 9,

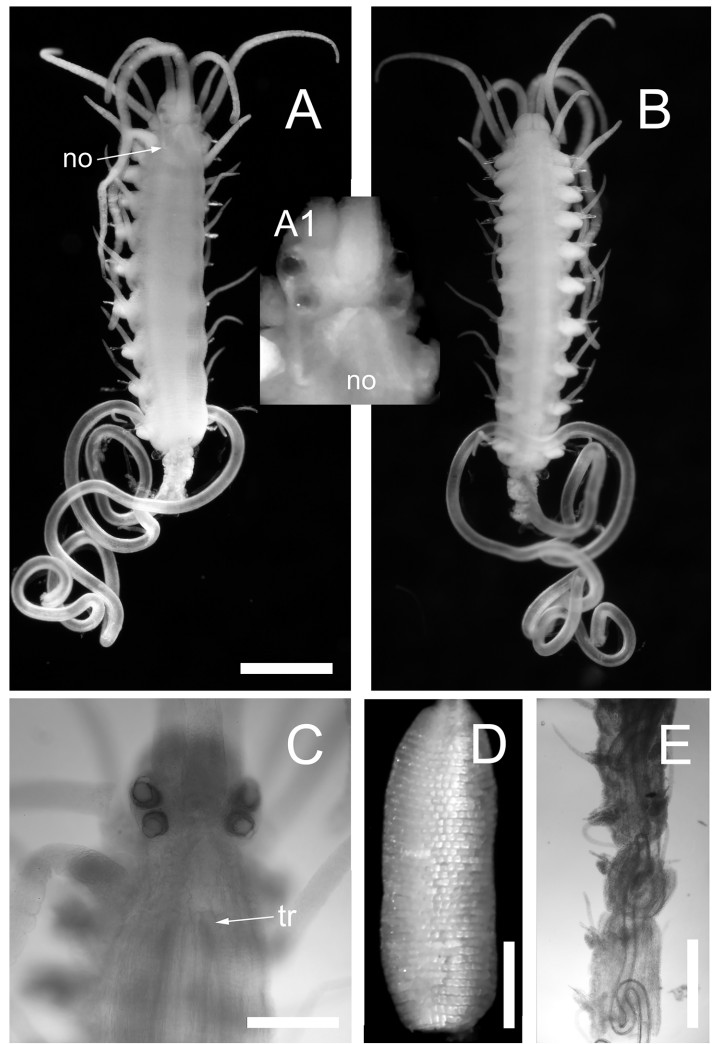

**Figure 3** *Proceraea exoryxae* **sp. nov.** (A) Anterior fragment, dorsal view [MNCN 16.01/17719], and detail of the head from the same specimen (A1). (B) Anterior fragment, ventral view [MNCN 16.01/17719]. Body is broken after chaetiger 10, exposing the pharynx (A and B). (C) Anterior end, dorsal view [MNCN 16.01/17719]. (D) Proventricle [MNCN 16.01/17719]. (E) Pharynx sinuation in chaetigers 9–14, dorsal view [MNCN 16.01/17720]. no, nuchal organs; tr, position of trepan. Scale bars A, B, E = 0.5 mm, C, D = 0.2 mm.

more posterior chaetigers with compound chaetae only. Simple chaetae unidentate with rows of spines subdistally (Figs. 4B, 5A–5D and 5F). In anterior 4–5 chaetigers most simple chaetae with a proportionally long region distal to the swollen neck (Figs. 4B and 5B–5D), one or two of the inferior-most chaetae with a shorter region distal to the swollen neck (Figs. 4B, 5A, 5C and 5D), similar in appearance to the shafts of the compound chaetae found in later chaetigers. Starting from chaetigers 6–7 all simple chaetae (except for the bayonet chaeta) nearly identical to the shafts of the compound chaetae (Figs. 4B and 5E). Blades of compound chaetae serrated, with two large distal teeth, distal-most slightly smaller, becoming smaller to almost disappear in mid-body and posterior chaetigers, shafts with a swollen neck with rows of spines (Figs. 4B and 5E).

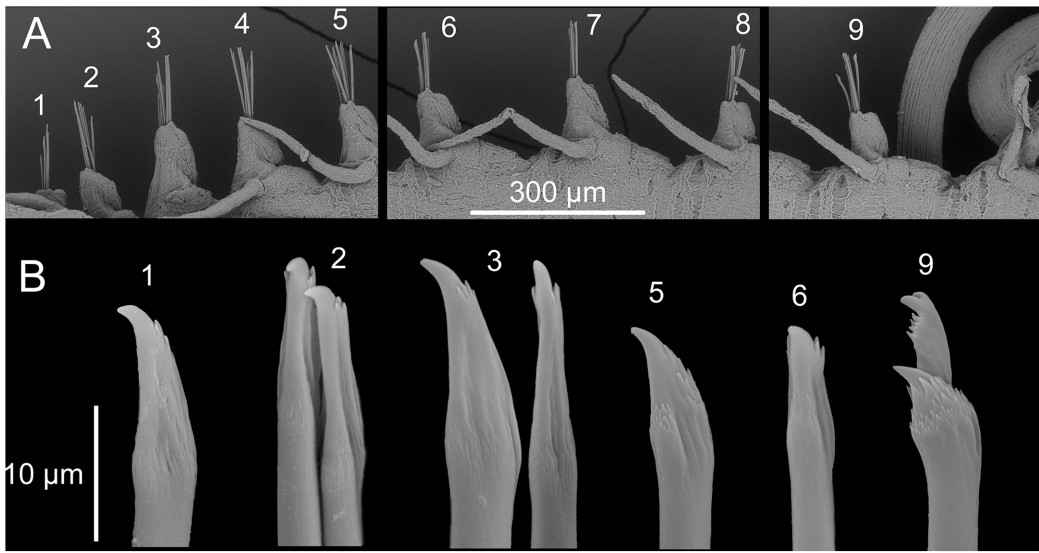

**Figure 4** *Proceraea exoryxae* sp. nov. SEM micrographs of chaetae structure [MNCN 16.01/17719]. (A) Chaetigers 1–9. (B) Chaetae: 1–3, simple chaetae with long region distal to the swollen neck from chaetigers 1–3; 5–6, simple chaetae with short region distal to the swollen neck from chaetigers 5 and 6; 9, compound chaetae from chaetiger 9.

Pharynx with several sinuations (Figs. 2C and 3E), mostly anterior to the proventricle, exact sinuation difficult to assess. Trepan at level of chaetiger 1–2 (Fig. 3C), with 33 teeth with one outer ring where one large tooth alternates with one medium-sized tricuspid tooth, and one inner ring with small teeth located just behind the large teeth (Figs. 6A–6C). Basal ring present, infradental spines absent. Proventricle as long as three segments in chaetiger 20–22 (uncertain observation, as the single specimen with proventricle still inside body looks distorted in this region), with 40–45 rows of square-shaped muscle cells ($n = 2$) (Figs. 2C and 3D).

**Distribution:** Known only from the type locality, Zeytouna Beach (Egyptian coast of the Red Sea).

**Etymology:** The specific epithet "*exoryxae*" derives from the term εξόρυξη, which means miner in Greek.

**Ecology:** *Proceraea exoryxae* sp. nov. was extremely rare. It was only found in one *Phallusia nigra* specimen, despite multiple successive collections of this ascidian in the same and other reefs during following years (*Kim et al., 2016*). The excavated galleries in which the new autolytine resided were visible only through the atrium wall (the internal surface of the tunic; Figs. 2B–2F), whereas the outside surface of the host tunic showed no signs of deformation, aside from the entrance openings of the galleries (Fig. 2E). The inner walls of the galleries were covered by a thin hyaline layer, apparently secreted by the worms. The wet mass of the individuals of *Phallusia nigra* collected in this reef ranged from 7.32 to 13.25 g and the specimen containing *Proceraea exoryxae* sp. nov. was 11.10 g. Two individuals of the amphipod *Leucothoe furina* (*Savigny, 1816*), a common

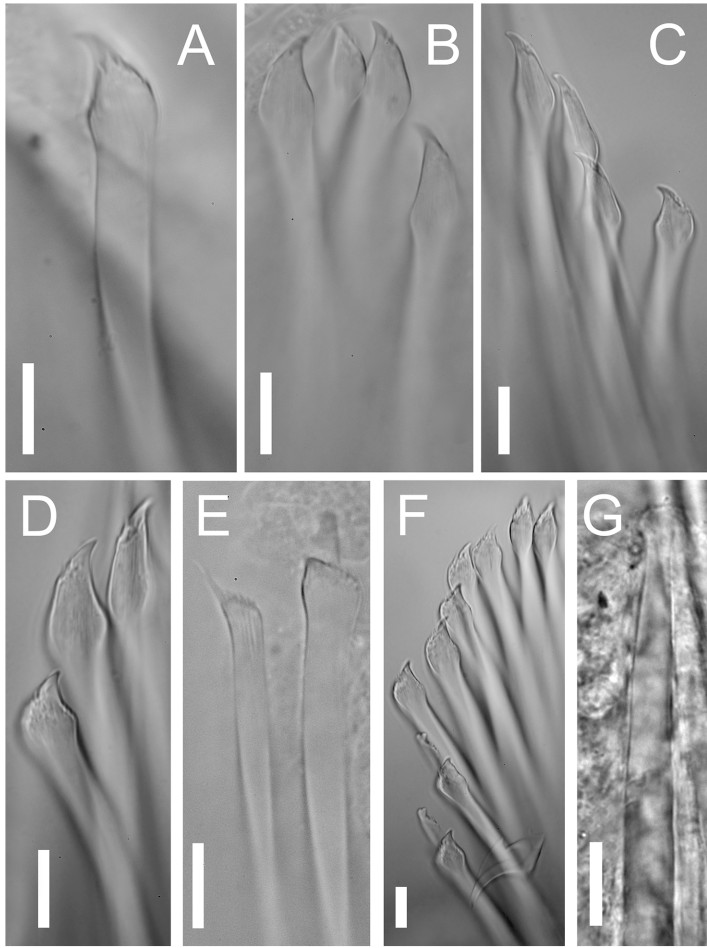

**Figure 5** ***Proceraea exoryxae* sp. nov. morphology of chaetae and aciculae.** (A) Inferior-most simple chaeta, chaetiger 1 [MNCN 16.01/17717]. (B) Simple chaetae, chaetiger 3 [MNCN 16.01/17717]. (C) Simple chaetae, chaetiger 4 [MNCN 16.01/17719]. (D) Simple chaetae, chaetiger 5 [MNCN 16.01/17719]. (E) Bayonet chaeta and compound chaeta, chaetiger 9 [MNCN 16.01/17717]. (F) Simple and compound chaetae, chaetiger 10 [MNCN 16.01/17719]. (G) Mid-body acicula [MNCN 16.01/17723]. Scale bars A–G = 0.1 mm.

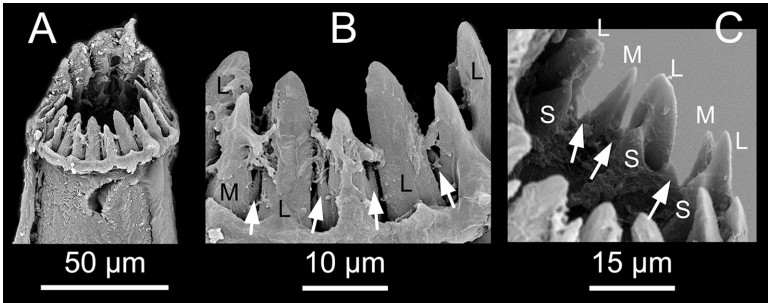

**Figure 6** ***Proceraea exoryxae* sp. nov. SEM micrographs of trepan structure.** (A) Whole view of the trepan (teeth on the back partly covered by tissue) [MNCN 16.01/17720]. (B) Large and medium, tricuspid teeth in external view. (C) Large, medium, tricuspid and small teeth in internal view. L, Large teeth; M, Medium, tricuspid teeth; S, small teeth; white arrows pointing on the lateral cusps.

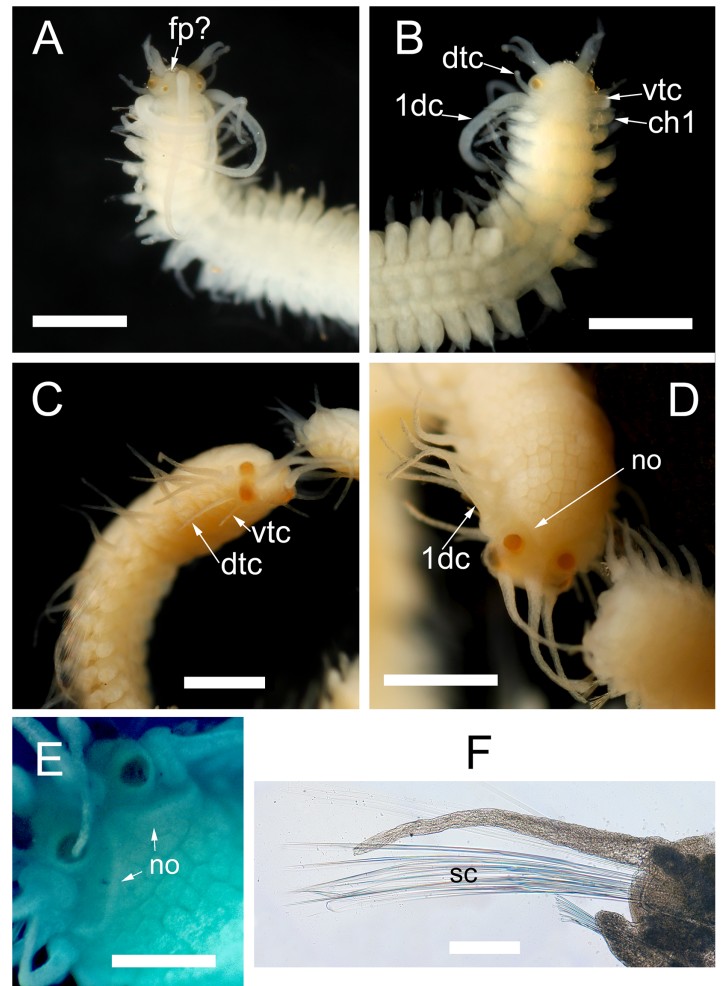

**Figure 7** *Proceraea exoryxae* **sp. nov. morphology of epitokes.** Anterior end of male stolon [MNCN 16.01/17721]: (A) dorsal view; (B) ventral view. Anterior end of female stolon [MNCN 16.01/17722]: (C) anteroventral view; (D) dorsal view; (E) detail of prostomium showing the nuchal organs (stained with methyl blue). (F) Mid-body parapodia of female stolon showing the swimming chaetae [MNCN 16.01/17722]. dtc, dorsal tentacular cirri; vtc, ventral tentacular cirri; 1dc, first dorsal cirri; ch1, chaetiger 1; fp, frontal process; no, nuchal organs; sc, swimming chaetae. Scale bars A–D = 0.5 mm, E, F = 100 μm.

associate of *Phallusia nigra* in the Egyptian Red Sea (*White, 2011*) were also found in the same host.

**Reproduction:** Probably with schizogamy, as several male and female stolons where found in the same galleries as the atokous forms (Figs. 2D–2F). Stolons were obtained detached from the corresponding stocks. However, they have bayonet and compound chaetae identical to those in the atokous forms, thus strengthening the hypothesis that they belong to *Proceraea exoryxae* sp. nov. Male and female stolons are described below.

**Morphology of the epitokous male.** Length 2.7 mm for 6+17 chaetigers in region a and b (*Nygren, 2004*), incomplete; width in region a 0.4 mm excluding parapodial lobes, in region b 0.7 mm including parapodial lobes. Exact color in vivo unknown, but either the ventral or the dorsal side of region b dark brownish, region a whitish, with diffuse darker

pigmentation (Fig. 2F). Preserved specimen whitish, without color markings, chaetiger 2–6 with paired yellowish sperm glands seen through the body wall (Fig. 7B). Prostomium rounded rectangular, wider than long, anterior margin convex. Four eyes with lenses, situated ventro-lateral and dorsal on prostomium, ventro-lateral pair larger (Figs. 7A and 7B). Palps absent. Nuchal organs not seen. Median antenna inserted medially on prostomium, reaching c. chaetiger 10; lateral bifid antennae, inserted on anterior margin, equal in length to prostomial width; basal part 1/3 of total length, outer ventral rami slightly longer and thinner than inner dorsal rami. Frontal processes possibly absent, or developing (seen as small protuberances on prostomium lateral to the median antenna) (Fig. 7A). Tentacular cirri 2 pairs (Fig. 7B), dorsal tentacular cirri, as long as 1/2 prostomial width, ventral tentacular cirri, 1/3 as long as dorsal pair. First dorsal cirri, equal in length to median antenna. Achaetous knobs absent. Cirri in region a reciprocally equal, equal in length to 1/2 body width excluding parapodial lobes, cirri in region b reciprocally equal, slightly shorter than cirri in region a. Short median ceratophore, and short cirrophores on first dorsal cirri, cirrophores otherwise absent. Median antenna, tentacular cirri, first dorsal cirri, and cirri in region a and b cylindrical. Parapodia in region a uniramous, neuropodial lobes rounded, parapodia in region b with developing notopodial lobes. Single neuropodial acicula in all chaetigers; 2 anterodorsal and 5 posteroventral notopodial aciculae in region b. Neuropodial fascicle with 7–8 compound chaetae and a single bayonet chaeta of the same types described for the atoke. Swimming chaetae absent, indicating a non-fully developed male stolon.

**Morphology of the epitokous female:** Length 5 mm for 6+27+9 chaetigers in region a, b and c (*Nygren, 2004*), incomplete; width in region a and c 0.6 mm excluding parapodial lobes, in region b 1 mm including parapodial lobes. Color of living individuals unknown. Preserved specimen yellowish, with body filled by eggs (Fig. 7D); color markings absent. Prostomium rounded rectangular, wider than long, anterior margin straight. Four eyes with lenses, situated ventro-lateral and dorsal on prostomium, ventro-lateral pair larger (Figs. 7C and 7D). Palps absent. Nuchal organs reaching beginning of chaetiger 1 (Figs. 7D and 7E). Median antenna inserted medially on prostomium, reaching c. chaetiger 5; lateral antennae inserted on anterior margin, about two third in length of median antenna. Tentacular cirri 2 pairs (Fig. 7C), dorsal tentacular cirri, as long as prostomial width, ventral tentacular cirri about 1/2 as long as dorsal pair. First dorsal cirri (Fig. 7D), equal in length to lateral antennae. Achaetous knobs absent. Cirri in region a reciprocally equal, slightly shorter than first dorsal cirri, equal in length to body width excluding parapodial lobes, cirri in region b reciprocally equal, slightly longer than cirri in region a, cirri in region c reciprocally equal, slightly shorter than cirri in region a. Ceratophores on median and lateral antennae, cirrophores present on all dorsal cirri, but tentacular cirri without cirrophores. Antennae, tentacular cirri, and dorsal cirri cylindrical. Parapodia in region a uniramous, neuropodial lobes rounded, parapodia in region b with additional notopodial lobes. Single neuropodial acicula in all chaetigers; 2–3 anterodorsal and 6–7 posteroventral notopodial aciculae in region b. Neuropodial fascicle with 7–8 compound chaetae and a single bayonet chaeta of the same types

described for the atokous form. Notopodial chaetal fascicle with 20–25 long and thin swimming chaetae (Fig. 7F).

## DISCUSSION

### Taxonomic remarks

The combination of morphological features in *Proceraea exoryxae* sp. nov. makes it difficult to place it in any specific genus within the Autolytinae. The thick type of bayonet chaeta, distally denticulated, and the presence of cirrophores only on anterior-most cirri indicate, however, that the new species is a member of the tribe Procerini. Accordingly, these morphological characters are not found in either of the two other main groups (Autolytini and *Epigamia*).

We place the new species in the genus *Proceraea* even though simple chaetae (apart from bayonet chaetae) are not found in any described member. We base our decision on the fact that the trepan teeth in *Proceraea exoryxae* sp. nov. are arranged in more than one ring, which is only found in *Proceraea* among Procerini. The observation of the trepan rings is clear under SEM, despite part of the dissected pharynx being still covered by tissue. The presence of simple chaetae in a restricted number of anterior chaetigers is a feature shared only with some members of *Procerastea* and *Imajimaea* among the Autolytinae, which differ in having trepans with a single ring, instead of separate rings as in *Proceraea exoryxae* sp. nov. Moreover, simple chaetae in *Proceraea exoryxae* sp. nov. differ from those in these two genera in that there seems to be two types. The first one (Figs. 4B(1–3)) has a peculiar morphology with an enlarged, hooked tip surrounded by a distal half crown of small denticles. In the second type, the hooked tip progressively reduces its length (e.g., Fig. 4(5–6)) to finally acquire a shape almost indistinguishable from the blades of compound chaetae (Fig. 4(9)). Only the first type of chaeta is present in the first chaetigers, and its number is progressively reduced to disappear around chaetiger 9–10. Conversely the second type progressively increased in number to be finally replaced by compound chaetae around chaetiger 10. At this level, it is almost impossible to distinguish between a simple chaeta and a compound one that has lost the blade. However, in mid-body and posterior segments, the presence of compound chaetae without blades is extremely rare. This, together with the fact that there is an antero-posterior gradation in tip length in the second type of simple chaetae is evidence that they are actually simple chaetae instead of compound ones without blade.

Further, all members of *Procerastea* have thick, distally dilated, bayonet chaetae and have dorsal cirri only on the first chaetiger, while *Proceraea exoryxae* sp. nov. has thick bayonet chaetae not distally dilated and dorsal cirri on all segments. *Imajimaea*, on the other hand, shares the presence of dorsal cirri on all its segments, except for *Imajimaea draculai* that lacks dorsal cirri on chaetigers 2–5. However, all species of *Imajimaea* have thin, subdistally denticulated, bayonet chaetae, and not thick bayonet chaetae, distally denticulated as in *Proceraea exoryxae* sp. nov.

Assuming that the assignment of the two stolons to this new species is correct, this may also shed some light on its taxonomic relationships. The type of stolon with six chaetigers in region a, two pairs of tentacular cirri, and no achaetous knobs is found in

*Virchowia clavata* Langerhans, 1879, *Virchowia pectinans* (Hartmann-Schröder, 1983), *Autolytus emertoni* Verrill, 1881, *Procerastea halleziana Malaquin, 1893, Procerastea nematodes* Langerhans, 1884 *and Proceraea picta* Ehlers, 1864 (*Nygren, 2004*). The information on the stolons of *Virchowia clavata, Procerastea* spp. and *Proceraea picta* is based on the literature only, but even in the species where the stolons are thoroughly described and illustrated, the achaetous knobs are not detailed. All other species assigned to *Proceraea* for which stolons are known, i.e., *P. cornuta* (*Agassiz, 1862*), *P. fasciata Bosc, 1802, P. hanssoni Nygren, 2004, P. nigropunctata* Nygren & Gidholm, 2001, *P. okadai* (Imajima, 1966), and *P. prismatica* (*Müller, 1776*), are equipped with achaetous knobs ventral to the first dorsal cirri. It is important to note that molecular phylogenetic studies have found the genus *Proceraea* to be paraphyletic without the inclusion of e.g., *Virchowia* and *Procerastea* (*Nygren et al., 2010*). *Proceraea picta* and close relatives are found as a sister group to a clade where the *Proceraea* having stolons with achaetous knobs constitute the first subclade, and *Virchowia, Procerastea* and other genera, whose known stolons lack achaetous knobs, constitute the second subclade. This indicates that having stolons with achaetous knobs is the derived state, while the lack of achetous knobs is plesiomorphic. A revision of *Proceraea* is clearly needed, but beyond the scope of this paper. Until then, we consider a generic assignment to *Proceraea* for this new species to be the best option.

## Autolytinid symbionts

Despite the hundreds of symbiotic polychaete species known, including >80 species considered parasitic, parasitism is relatively rare in this taxon when compared to other lifestyles (*Martin & Britayev, 1998*; *Britayev & Lyskin, 2002*; *Britayev et al., 2014*). Most recorded symbiotic associations between polychaetes and invertebrates involve sponge, cnidarian, mollusc, or echinoderm hosts, but a few mention ascidians. Some previous reports of polychaetes living among ascidians came from dredged or scraped-quadrat samples, which are usually inadequate to determine symbiont–host associations because they result in specimen mixtures, while soft-bodied animals, like tunicates or annelids, are often greatly damaged. In this context, the term "associated with" most often refers to spatially co-occurring specimens. Nonetheless, high densities of polychaetes, including syllids, are known to occur among the fauna associated with particular benthic tunicates (*Allen, 1915*; *Fielding, Weerts & Forbes, 1994*; *Cerdá & Castilla, 2001*; *Fiore & Jutte, 2010*; *Sepúlveda et al., 2015*). Polychaetes were dominant on intertidal (but not subtidal) beds of *Pyura stolonifera* (*Heller, 1878*), the second most abundant group in beds of the congeneric *Pyura praeputialis* (*Heller, 1878*) (*Fielding, Weerts & Forbes, 1994*; *Cerdá & Castilla, 2001*), and constituted 28% of the fauna associated with didemnid ascidians (*Fiore & Jutte, 2010*). These reports, however, largely refer to animals living in the sediments accumulated in the crevices among ascidian aggregates and, thus, there is no reason to suspect true symbiotic interactions. Similarly, intraspecific variation in growth form of *Pyura chilensis Molina, 1782* has been documented in response to the presence of chaetopterid polychaete tubes in the assemblage (*Sepúlveda et al., 2015*), but this was interpreted as a density-dependent phenomenon not related to symbiosis.

**Table 1  List of known autolytinid syllids reported as symbionts.**

| Symbiont | | Host | References |
|---|---|---|---|
| *Proceraea* sp. | Cn | *Abietinaria turgida* (*Clarke, 1877*) | *Britayev & San Martín (2001)* |
| | Cn | *Orthopyxis integra* (*Macgillivray, 1842*) | T. A. Britayev, 2015, personal communication |
| *Imajimaea draculai* (*San Martín & López, 2002*) | Cn | *Funiculina quadrangularis* (*Pallas, 1766*) | *Nygren & Pleijel (2010)* |
| *Myrianida piningera* (*Montagu, 1808*) | Tu | *Phallusia mammillata* (*Cuvier, 1815*) | *Okada (1935)*, *Spooner, Wilson & Trebble (1957)* |
| | Tu | *Ascidiella aspersa* (O. F. *Müller, 1776*) | *Okada (1935)*, *Spooner, Wilson & Trebble (1957)* |
| *Proceraea cornuta* (*Agassiz, 1862*) | Cn | Unidentified hydroid | *Pettibone (1963)* |
| | Cn | Unidentified Coral | *Gardiner (1976)* |
| *Procerastea halleziana* *Malaquin, 1893* | Cn | *Ectopleura crocea* (*Agassiz, 1862*) | *Genzano & San Martín (2002)* |
| | Cn | *Coryne eximia* *Allman, 1859* | *Allen (1915, 1923)*, *Spooner, Wilson & Trebble (1957)*, *Alós (1989)* |
| | Cn | *Tubularia indivisa* *Linnaeus, 1758* | *Caullery (1925)*, *Spooner, Wilson & Trebble (1957)* |
| *Proceraea penetrans* (Wright and Woodwick, 1977) | Cn | *Stylaster californicus* (*Verrill, 1869*) | *Wright & Woodwick (1977)* |
| *Proceraea madeirensis* *Nygren, 2004* | Cn | *Eudendrium carneum* *Clarke, 1882* | E. Cruz-Rivera, 1991, personal observations |
| *Pachyprocerastea hydrozoicola* (*Hartmann-Schröder, 1992*) | Cn | *Pseudosolanderia* sp. | *Hartmann-Schröder (1992)* |
| *Procerastea parasimpliseta* *Hartmann-Schröder, 1992* | Cn | *Pseudosolanderia* sp. | *Hartmann-Schröder (1992)* |

**Note:**
Cn, Cnidarians; Tu, tunicates.

Other studies have documented serendipitous observations obtained while searching for other ascidian associates. For example, in his monograph on ascidian-associated copepods, *Illg (1958)* reported unidentified polychaetes from the atria and branchial sacs of dredged ascidians. Similarly, *Monniot, (1990)* reported ten unidentified Syllidae from the branchial sac of *Microcosmus anchylodeirus* Traustedt, 1883. In summary, information on the nature of polychaete-ascidian relationships remains very scarce. Most reports of polychaetes (and syllids in particular) do not refer to individuals "living in association with" tunicates (which would imply some degree of specialization and thus suggest any type of symbiotic interaction). In fact, there is only one previous report specifically referring to a symbiosis, in which another autolytine, *Myrianida pinnigera*, was found living in association with *Ascidiella aspersa* and *Phallusia mammilata* (Table 1). Two decades later, *Spooner, Wilson & Trebble (1957)* stated that this species feeds on the body fluids of these and other ascidians from the British southern coast. While intriguing, this interaction has never been quantitatively evaluated and the evidence for this specialized trophic mode is still unclear. But if so, *Proceraea exoryxae* sp. nov. is the second known polychaete, and the second autolytine too, living in symbiosis with ascidians.

In addition to these two species of tunicate associates, eight more autolytines have been previously reported as living in symbiosis with other invertebrates, all them cnidarians (Table 1). Most of them are considered parasites, although only some are sufficiently studied to be clearly defined as such (*Martin & Britayev, 1998*). Among the best

documented, *Proceraea penetrans* (*Wright & Woodwick, 1977*) induces galls on its hydrocoral hosts, while *Proceraea* sp. modifies the theca of polyps in its hydroid hosts in order to live inside, probably feeding on the polyps themselves (*Britayev, San Martín & Sheiko, 1998*; *Britayev & San Martín, 2001*).

### *Proceraea exoryxae* sp. nov. as a miner

The association of *Proceraea exoryxae* sp. nov. with *Phallusia nigra* appears to be extremely rare, as there was only one infested host among all those we examined. The presence of a polychaete inside the tunic of *Phallusia nigra* has not been reported in previous studies at the same and other reefs (*Kim et al., 2016*). As mentioned above, parasitism is an atypical phenomenon among polychaetes, but also parasitic species are, with a few exceptions, extremely rare. In fact, many symbiotic polychaetes are only known from a single specimen (or very few) found only once (*Martin & Britayev, 1998*). The reasons for this rarity are often unknown. We may speculate that the lack of dedicated studies may be the actual reason in many cases, although that seems unlikely for *Proceraea exoryxae* sp. nov., which was discovered during multi-year monitoring of the associated fauna of the host ascidian (*Kim et al., 2016*). We could also hypothesize that the parasite is a recent introduction from an unknown origin, but this also seems unlikely because the host is a Red Sea endemic ascidian (*Vandepas et al., 2015*) and specialist parasites would be expected to occur in areas where hosts have the longest evolutionary history. More reasonably, either the polychaete occurs only infrequently and is thus difficult to find, or its peculiar and hidden habitat may have caused it to be overlooked in previous studies. We can also not discard the possibility that the parasitic mode of life may be just a phase in the life cycle of the worm, possibly connected to reproduction, as inferred from the presence of epitokous forms among atokes. This would add a temporal component to the presence of *Proceraea exoryxae* sp. nov. inside *Phallusia nigra*, that would increase the difficulty in finding it.

Despite (and, maybe, due to) its rarity, *Proceraea exoryxae* sp. nov. is the first polychaete formally defined as miner and, certainly, the first of Autolytinae. We use the term mining as it is often used to describe insects that tunnel through the tissues of their plant hosts (*Connor & Taverner, 1997*; *Sinclair & Hughes, 2010*; *Mejaes, Poore & Thiel, 2015*), but also marine isopods tunneling seagrass leaves (*Brearley & Walker, 1995*). This is also the mechanism we suggest for the formation of the galleries in the *Phallusia nigra* tunic where *Proceraea exoryxae* sp. nov. was found. The rarity of the polychaete precluded a thorough assessment of the host-symbiont interaction although, as in the case of *M. pinnigera*, the new species possibly feeds on the tissues of the host ascidian. Nonetheless, it represents the first clear example of mechanical damage by a polychaete on an ascidian, and as such, we classify the interaction as a parasitic symbiosis (*Castro, 2015*). The defensive characteristics attributed to the *Phallusia nigra* tunic, which include the accumulation of vanadium and sulfuric acid, and their derived metabolites (*Stoecker, 1980*; *Hirose, Yamashiro & Mori, 2001*; *Pisut & Pawlik, 2002*; *Odate & Pawlik, 2007*), did not prevent infestation by *Proceraea exoryxae* sp. nov., while they have been suggested to prevent infestation by the bivalve *Musculus subpictus* (*Cantraine, 1835*) in

an population introduced in Panama (*Cañete & Rocha, 2013*). Because both the host and polychaete symbiont were likely at their native habitat, and because symbionts are often unaffected by host defensive metabolites, the new partnership here reported may imply a noticeable degree of specialization. The presence of epitokous forms certainly confirms that at least the first phases of the reproductive cycle of the species (i.e., stolon formation) occurred inside the galleries, which may be considered as an additional evidence of specialization. However, whether *Proceraea exoryxae* sp. nov. is an exclusive parasite of *Phallusia nigra* or infests other ascidians awaits further studies.

Although rare for polychaetes, many invertebrates are known to live in symbiotic associations with ascidians, including amphipods, shrimps, copepods, pinnotherid crabs, nemerteans and cnidarians (*Illg, 1958*; *Stock, 1967*; *Boxshall, 2005*; *Lambert, 2005*; *Monniot, 1990*; *Thiel, 2000*; *Baeza & Díaz-Valdés, 2011*; *White, 2011*; *Kim et al., 2016*). Most of these animals live in the branchial sac of the host and are often considered commensals, with the exception of some copepod taxa, which are largely classified as ectoparasites on this respiratory organ (*Illg, 1958*; *Stock, 1967*; *Boxshall, 2005*; *Kim et al., 2016*). In contrast, but perhaps not surprisingly, few animals have evolved to inhabit the ascidian tunic, which is often structurally tough, and may contain spicules, inorganic acids, concentrated vanadium, and a variety of defensive secondary chemicals (*Stoecker, 1980*; *Pisut & Pawlik, 2002*; *Joullié et al., 2003*; *Odate & Pawlik, 2007*; *Koplovitz et al., 2009*). Some mytilid mussels in the genera *Mytilimeria* and *Musculus* (=*Modiolarca*) are symbiotic bivalves that live completely embedded in the tunic of their ascidian host (*Say, 1822*; *White, 1949*; *Lambert, 2005*; *Morton & Dinesen, 2011*; *Cañete & Rocha, 2013*). Similarly, two species of amphipods in the genus *Polycheria* live by filter feeding from individual shallow pockets they excavate on the tunic of their host ascidians (*Skogsberg & Vansell, 1928*; *McClintock et al., 2009*). Recently, the parasitic copepod *Janstockia phallusiella Boxshall & Marchenkov, 2005* has been reported as living attached to the atrial wall of *Phallusia nigra* (*Kim et al., 2016*). None of these animals, however, produce a network of tunnels similar to that observed in the specimen of *Phallusia nigra* infested by *Proceraea exoryxae* sp. nov.

Among polychaetes several species are known to inhabit excavated galleries. Probably the best known are polydorid spionids, which include numerous species from different genera that burrow into calcareous substrates, including algae, crustacean carapaces, and mollusc shells. Some of them are simple borers, but others are well-known commensals and parasites, sometimes being even considered as pests when they infest species of commercial interest (*Martin & Britayev, 1998*). Although less diverse, similar habits are also present among cirratulids and sabellids, the latter being also able to infest fresh water invertebrates (*Martin & Britayev, 1998*). Polychaetes are also known to excavate galleries in seagrasses (*Guidetti, 2000*; *Gambi, van Tussenbroek & Brearley, 2003*), cnidarians (*Martin et al., 2002*; *Cairns & Bayer, 2008*; *Cairns, 2006*, *2009*, *2011*, *2012*; *Mueller et al., 2013*; *Britayev et al., 2014*; *Molodtsova, Britayev & Martin, 2016*) and sponges (see *Lattig & Martin, 2011* and references herein). Seagrass associated polychaetes are mainly detritivores that bore into the dead sheath tissues (*Gambi, van Tussenbroek & Brearley, 2003*), but their galleries are also present in living meristems and leaves that have

been reported as "mined" tissues (*Guidetti, 2000*). Cnidarian associates (e.g., polynoids, eunicids, syllids) may inhabit depressions in the host skeleton that are usually covered by overgrowing host tissues and/or skeleton to form tunnels or galleries, presumably as a reaction to the symbionts' presence (*Britayev et al., 2014*). A particular case is that of *Haplosyllis anthogorgicola Utinomi, 1956*, which excavates a network of galleries inside the soft tissues of its host gorgonian. Host tissue overgrowths are limited to small tube-like protuberances at the gallery exits, from where the worms supposedly feed by stealing food from the nearby host polyps (*Martin et al., 2002*). Polychaete sponge borers are mainly syllids (e.g., *Haplosyllis, Haplosyllides*), which may either inhabit the aquiferous channels of the sponge or excavate their own galleries inside the host tissues (*Martin & Britayev, 1998*; *Martin, Aguado & Britayev, 2009*; *Lattig & Martin, 2011*).

When observing the tunic of the Red Sea specimen of *Phallusia nigra* we did not detected traces of external overgrowths associated to the gallery openings and, when dissecting the excavated galleries, we did not find any induced malformations or defined cavities (like cysts, galls or blisters). Conversely, there was a thin, translucent layer covering the galleries. Likely, this was an inner lining secreted by the worms to cover the tunnel walls, possibly made in a similar fashion as the hyaline tubes that some autolytines build to remain attached to their host cnidarians (*Molodtsova, Britayev & Martin, 2016*). At present, the mechanics of excavating tunnels by *Proceraea exoryxae* sp. nov. are unknown, but the typical syllid feeding structures (i.e., trepan, evaginable pharynx and sucking proventricle) seem to be a perfect combination enabling *Proceraea exoryxae* sp. nov. for this particular task.

In addition to possible affectations to host fitness, the parasitic mode of life attributed to *Proceraea exoryxae* sp. nov. may also be relevant for coastal management. Being native from the Red Sea, *Phallusia nigra* has been introduced worldwide in tropical and sub-tropical ecosystems (*Shenkar, 2012*; *Vandepas et al., 2015*) where, as many other tunicates (*Zhan et al., 2015*), it has the potential of becoming invasive. Accordingly, three interesting questions remain open for further studies: (1) whether the parasitic *Proceraea exoryxae* sp. nov. may be (or has already been) introduced together with the ascidian, (2) whether it may contribute to control the spreading of *Phallusia nigra* in non-native regions, and (3) whether it may switch its host to infest, and thus cause damage, to native ascidians in the regions were the Red Sea host/parasite partnership has been introduced. In combination with molecular tools to trace the origin of an introduced species, the existence of a specialized parasite known only from the native host population may also help assess whether the host species has been introduced directly from this native population or indirectly from an already introduced population (*MacKenzie, 1993*, *2002*; *Catalano et al., 2014*). Nevertheless, the actual relevance of the association may be obscured by its rarity and, thus, will certainly rely on a future confirmation of its actual prevalence, as well as on the assessment of spatial and temporal extension of the infestation.

## ACKNOWLEDGEMENTS

EC-R acknowledges the Department of Biological Sciences, University of the Virgin Islands for time release during the preparation of this article. Hussa Al Ajeer, Ali Fahmi,

Maricel Flores-Díaz, and Tamer Hafez, helped with specimen collection and ascidian dissections. We thank the staff of the J. D. Gerhart Field Station for support during our fieldwork and Dr X. Turon from the CEAB–CSIC for his advice on tunicate associates. María García, from the SEM service at the CEAB–CSIC, kindly helped with the SEM observations. This paper is contribution #175 from the Center for Marine and Environmental Sciences, University of the Virgin Islands (EC-R), and a contribution of DM to the research project MarSymBiomics and the Consolidated Research Group on Marine Benthic Ecology.

### Funding

EC-R was supported by a Faculty Improvement Grant from the American University in Cairo. DM was supported by research project MarSymBiomics (grant number CTM2013-43287-P) funded by the Ministerio de Educación, Cultura y Deporte, and by the Consolidated Research Group on Marine Benthic Ecology (grant number 2014SGR120) of the Agència de Gestió d'Ajuts Universitaris i de Recerca of the Generalitat de Catalunya. The funders had no role in study design, data collection and analysis, decision to publish, or preparation of the manuscript.

### Grant Disclosures

The following grant information was disclosed by the authors:
American University in Cairo.
MarSymBiomics: CTM2013-43287-P.
Ministerio de Educación, Cultura y Deporte.
Consolidated Research Group on Marine Benthic Ecology, Agència de Gestió d'Ajuts Universitaris i de Recerca of the Generalitat de Catalunya: 2014SGR120.

### Competing Interests

The authors declare that they have no competing interests.

### Author Contributions

- Daniel Martin conceived and designed the experiments, performed the experiments, analyzed the data, contributed reagents/materials/analysis tools, wrote the paper, prepared figures and/or tables and reviewed drafts of the paper.
- Arne Nygren conceived and designed the experiments, performed the experiments, analyzed the data, contributed reagents/materials/analysis tools, wrote the paper, prepared figures and/or tables and reviewed drafts of the paper.
- Edwin Cruz-Rivera conceived and designed the experiments, performed the experiments, analyzed the data, contributed reagents/materials/analysis tools, wrote the paper, prepared figures and/or tables, reviewed drafts of the paper and collected the specimens.

## Data Availability

Specimen vouchers were deposited at the Museo Nacional de Ciencias Naturales of Madrid, Spain (MNCN). Microscope photos are included in the manuscript.

## New Species Registration

The following information was supplied regarding the registration of a newly described species:

Species name: urn:lsid:zoobank.org:act:34373CE6-A0D4-488D-B4A5-12CF4E103504

Publication LSID: urn:lsid:zoobank.org:pub:685CB1C2-CB5B-4A87-9CD7-C04BFFDE03B4

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
