# Peer review of "Proceraea exoryxae sp. nov. (Annelida, Syllidae, Autolytinae), the first known polychaete miner tunneling into the tunic of an ascidian"

_PeerJ, doi:10.7717/peerj.3374_

## Round 0.1 · original submission · Minor Revisions

Dear Author, thanks for submitting your interesting ms to PeerJ.

The reviewers have concurred that the ms is of high quality and requires only minor modification to be suitable for publication. However, interestingly, all three of us have picked on different things, which we consider most important, and with which I concur: Rev 1: that you add information on infestation rate in relation to 1. symbiotic specialisations, and 2. coastal management; Rev 2: that some consideration is given to the possibility that only one type of chaeta exists (the second being a damaged version of the first); and Editor: that some of the images be improved where possible, particularly Figure 3. Overall though, an interesting, well crafted ms. I look forward to your response and revised ms,

All the best, Chris

·

Basic reporting

The paper is very clear, unambiguous. Professional english is also used.
The authors give a proper introduction and background. The literature given is relevant and well referenced.
All the figures and the table are also relevant, well labelled and described, and they help to understand the content of the paper

Experimental design

everything described with enough detail, although I miss a comment about where was the material deposited (line 113 in the attached pdf).
Some other suggestions are also included in the attached pdf.

Validity of the findings

The paper is very interesting and I have enjoyed a lot its reading.
However, although the authors give several and detail explanations about the taxonomy of the new species, and the implications of the host-symbiont relationship, the paper lacks a discussion related to the fact that they only found syllids within a unique specimen of P. nigra.
Given that the authors say that "the new partnership here reported may be a well-established relationship with a noticeable degree of specialisation" and also they suggest that this new association "may also be relevant for coastal management", I think it is important and necessary to explain that part.

·

Basic reporting

This is a very interesting, important and clearly written paper describing parasitic association between new species of syllid polychaete and its host ascidian for the first time. Introduction is well organized and provides enough background to demonstrate the novelty of the paper. The structure of the manuscript conforms to PeerJ standards. Figures are relevant, of a high quality, and well labelled and described. Literature is well referenced and relevant.

Experimental design

This original and primary paper is within the scope of the journal. Research questions are well defined, relevant and addressed to cover the gaps in knowledge on the rare phenomenon of parasitic relationships involving polychaetes. The methods employed are sufficient to obtain responses on questions arisen. They are relevant, described with sufficient details, and informative enough to replicate.

Validity of the findings

Species description is very carefully and professionally done and includes not only morphology of adults, but reproductive stages as well. The authors demonstrated the mining activity of polychaetes in the ascidian tunic, which has a particular interest due to the very strong defensive characteristics attributed to the tunic, including the accumulation of vanadium and sulfuric acid and their metabolites. The manuscript contains also a brief overview of other organisms living in symbiotic associations with ascidians to underline that just a few of them are able to colonize structurally the tough and chemically protected tunic, and also an overview of all autolytinid polychaetes known as symbionts, which is particularly relevant in the general context of the manuscript.

Additional comments

I have no any special comments to the description except one. The second type of simple chaetae found by the authors in the some of the anterior segments of the worms (from 4 to 13) are very similar in shape to the shafts of compound chaetae of the same worm after lost of the distal blades. I suggest the authors have to explain why they consider them as a different type of chaetae. This explanation is especially relevant taking into account that simple chaetae are not characteristic to other Proceraea species, and seem to be required independently of the fact that the worms have another type of simple chaetae in the anterior-most chaetigers, with a clearly different and singular morphology. I am looking forward to see this in print after minor revision.

---

## Round 0.2 · accepted · Accept

Dear Edwin, Thanks for the revised ms, and for responding to the suggestions of the reviewers. The changes made as a result of the reviewers suggestions have improved the quality of the ms. Can I suggest two more edits related to the new text, l. 357 'indistinguishable', and l 364, a rewording option ... 'is evidence that they are actually ....'. I look forward to seeing your ms published.
Best, Chris